

# Freeze-thaw decellularization of the trabecular meshwork in an *ex vivo* eye perfusion model

Yalong Dang[1], Susannah Waxman[1], Chao Wang[1,2], Adrianna Jensen[1], Ralitsa T. Loewen[1], Richard A. Bilonick[1] and Nils A. Loewen[1]

[1] Department of Ophthalmology, University of Pittsburgh, Pittsburgh, PA United States of America
[2] Department of Ophthalmology, Xiangya Hospital, Central South University, Changsha, China

Corresponding author
Nils A. Loewen,
loewen.nils@gmail.com

## ABSTRACT

**Objective**. The trabecular meshwork (TM) is the primary substrate of outflow resistance in glaucomatous eyes. Repopulating diseased TM with fresh, functional TM cells might be a viable therapeutic approach. Decellularized TM scaffolds have previously been produced by ablating cells with suicide gene therapy or saponin, which risks incomplete cell removal or dissolution of the extracellular matrix, respectively. We hypothesized that improved trabecular meshwork cell ablation would result from freeze-thaw cycles compared to chemical treatment.

**Materials and Methods**. We obtained 24 porcine eyes from a local abattoir, dissected and mounted them in an anterior segment perfusion within two hours of sacrifice. Intraocular pressure (IOP) was recorded continuously by a pressure transducer system. After 72 h of IOP stabilization, eight eyes were assigned to freeze-thaw (F) ablation ($-80\,^{\circ}\text{C} \times 2$), to 0.02% saponin (S) treatment, or the control group (C), respectively. The TM was transduced with an eGFP expressing feline immunodeficiency viral (FIV) vector and tracked via fluorescent microscopy to confirm ablation. Following treatment, the eyes were perfused with standard tissue culture media for 180 h. TM histology was assessed by hematoxylin and eosin staining. TM viability was evaluated by a calcein AM/propidium iodide (PI) assay. The TM extracellular matrix was stained with Picro Sirius Red. We measured IOP and modeled it with a linear mixed effects model using a B-spline function of time with five degrees of freedom.

**Results**. F and S experienced a similar IOP reduction of 30% from baseline ($P = 0.64$). IOP reduction of about 30% occurred in F within 24 h and in S within 48 h. Live visualization of eGFP demonstrated that F conferred a complete ablation of all TM cells and only a partial ablation in S. Histological analysis and Picro Sirius staining confirmed that no TM cells survived in F while the extracellular matrix remained. The viability assay showed very low PI and no calcein staining in F in contrast to many PI-labeled, dead TM cells and calcein-labeled viable TM cells in S.

**Conclusion**. We developed a rapid TM ablation method that uses cyclic freezing that is free of biological or chemical agents and able to produce a decellularized TM scaffold with preserved TM extracellular matrix in an organotypic perfusion culture.

## INTRODUCTION

The trabecular meshwork (TM) is the primary substrate of outflow resistance in normal and glaucomatous eyes. Recent studies suggested not only low TM cellularity (*Alvarado, Murphy & Juster, 1984*; *Baleriola et al., 2008*), but also TM cytoskeleton and phagocytosis changes in primary open angle glaucoma (*Clark et al., 1995*; *Fatma et al., 2009*; *Izzotti et al., 2010*; *Saccà, Pulliero & Izzotti, 2015*; *Peters et al., 2015*; *Micera et al., 2016*). Repopulating diseased TM with fresh, functional TM cells has been shown to restore homeostasis of normal outflow and thus might represent a novel therapeutic breakthrough (*Du et al., 2013*; *Abu-Hassan et al., 2015*; *Yun et al., 2016*; *Zhu et al., 2016*).

For TM cell transplantation studies, preserving the structure and the extracellular matrix is desirable to provide a natural transplantation environment. Eliminating or reducing the number of host TM cells is also useful. In a recent study, an ex vivo 3D bioengineered TM scaffold repopulated by human primary TM cells was developed, but without the distinct layers of juxtacanalicular, corneoscleral and uveal TM (*Torrejon et al., 2016*). Transgenic (Tg-MYOC Y437H (*Zhu et al., 2016*) and laser photocoagulation mouse models (*Yun et al., 2014*) have also been used or proposed for TM transplantation, respectively. However, the anatomy of the rodent outflow tract has only a limited number of TM cell layers (three to four) compared to that of humans (*Ko & Tan, 2013*). Porcine eyes share many features that are similar to human eyes, including size, structure, intraocular pressure (IOP), the outflow pattern (*Sanchez et al., 2011*; *Loewen et al., 2016e*; *Loewen et al., 2016a*) and a large trabecular meshwork that guards the angular aqueous plexus (*Tripathi, 1971*) which has Schlemm's canal-like segments (*Suárez & Vecino, 2006*). The presence of biochemical glaucoma markers in the pig (*Suárez & Vecino, 2006*), genomic similarities to humans that rival that of mice (*Ensembl, 2015*; *Groenen et al., 2012*; *Flicek et al., 2014*) and microphysiological properties such as giant vacuole formation by Schlemm's canal endothelium (*McMenamin & Steptoe, 1991*) suggest pig eyes as glaucoma research models (*Ruiz-Ederra et al., 2005*).

*Abu-Hassan et al. (2015)* used saponin as an elegant way to induce a glaucoma-like dysfunction and cell loss in the TM of pig eyes with a $36\% \pm 9\%$ cell count reduction at 10 min. Saponins are a mixed group of plant derived, steroid and terpenoid glycosides that are used as detergents. The impact on remaining host and transplanted donor TM cells as well as on the ECM is not known. To address these concerns, we developed a chemical-free, freeze-thaw method to produce a decellularized TM scaffold. Together with our anterior segment perfusion system (*Loewen et al., 2016e*), this scaffold model can be used for cell transplantation, allowing real-time TM visualization and IOP measurement.

## MATERIALS AND METHODS

### Study design

Pig eyes were obtained from a local abattoir and prepared for culture within 2 h of death. Twenty-four eyes were assigned to three groups: freeze-thaw (F, $n = 8$), saponin (S, $n = 8$) and control (C, $n = 8$). This number was chosen based on past power calculations and the maximum number that could be perfused simultaneously thereby minimizing the

variability with same group experiments with our setup (*Loewen et al., 2016e*; *Loewen et al., 2016a*). Anterior segment perfusion cultures were allowed to stabilize for 72 h before being subjected to freeze-thaw cycles or to saponin supplemented media, respectively. The intraocular pressure (IOP) was recorded continuously by a pressure transducer system (Physiological Pressure Transducer, SP844; MEMSCAP, Skoppum, Norway) (*Loewen et al., 2016e*; *Loewen et al., 2016a*). Anterior segment cultures were perfused for another 180 h. Two additional eyes per ablation method group were transduced with eGFP expressing feline immunodeficiency viral vectors and subjected to the same ablation methods as used in the experimental groups. Expression of eGFP was monitored and compared. Two eyes per group were randomly selected for TM viability assays, histological analysis, and ECM assessment.

## Preparation of porcine anterior segments and perfusion system

After removing extraocular tissues, freshly enucleated porcine eyes from a local abattoir (Thoma Meat Market, Saxonburg, PA, USA) were placed into a 5% povidone-iodine solution (NC9771653; Fisher Scientific, Waltham, MA, USA) for 3 min and rinsed three times with phosphate-buffered saline (PBS). Eyes were hemisected 7 mm posterior and parallel to the limbus and the lens, ciliary body, and iris were carefully removed. Anterior segments were again washed with PBS three times and mounted in anterior segment perfusion dishes. Phenol-free DMEM media (SH30284; HyClone, GE Healthcare, UK)) supplemented with 1% fetal bovine serum and 1% antibiotic-antimycotic (15240062; Thermo Fisher Scientific, Waltham, MA, USA) was continuously pumped into the anterior chambers at a constant infusion rate of 3 microliters per minute. After calibration of the pressure transducers, the IOP was recorded at 2-minute intervals.

## TM ablation by freeze-thaw cycles or 0.02% saponin

After allowing the eyes to stabilize for 72 h, the groups were subjected to freeze-thaw cycles or 0.02% saponin, respectively. In the freeze-thaw group, anterior segments were exposed to −80 °C for 2 h, then thawed at room temperature for 1 h. After two cycles, anterior segments were reconnected to the perfusion system. In the saponin group, the regular perfusion media was replaced with 0.02% saponin supplemented media for 15 min, then exchanged for the normal perfusion media in a 37 °C incubator as described before (*Abu-Hassan et al., 2015*).

## TM transduction and visualization

Feline immunodeficiency viral vectors expressing eGFP were generated by transient cotransfection of envelope plasmid pMD.G, packaging plasmid pFP93, and gene-transfer plasmid encoding eGFP and neomycin resistance GINSIN (*Saenz et al., 2007*; *Oatts et al., 2013*; *Zhang et al., 2014*) using a polyethylenimine method (*Loewen et al., 2016e*). Supernatant from 293T transfected cells containing GINSIN vector was harvested at two, four and six days after transfection, then concentrated by ultracentrifugation. $1 \times 10^7$ transducing units of GINSIN were injected into the anterior chambers. eGFP expression was visualized through the bottom of the culture dish using a dissecting microscope equipped for epifluorescence (SZX16; Olympus, Tokyo, Japan).

### TM viability

TM viability was evaluated by calcein acetoxymethyl (calcein AM) and propidium iodide (PI) co-labelling (*Gonzalez Jr, Hamm-Alvarez & Tan, 2013*; *Dang et al., 2017a*). After 180 h, the anterior segments were collected and washed with PBS three times. The limbus with the TM was dissected and incubated with calcein AM (0.3 μM, C1430; Thermo Fisher, Waltham, MA, USA) and PI (1 μg/ml, P1304MP; Thermo Fisher, Waltham, MA, USA) for 30 min at 37 °C. After three additional PBS washes, the TM was flat-mounted, and imaged under an upright laser scanning confocal microscope at 400-fold magnification (BX61; Olympus, Tokyo, Japan).

### TM histology

TM samples obtained from at least two separate quadrants per eye were dissected and fixed with 4% paraformaldehyde in PBS for 24 h. After rinsing them three times in PBS, they were embedded in paraffin, sectioned at 6-micron thickness and stained with hematoxylin and eosin.

### TM-ECM assessment

The TM-ECM was assessed by a Picro Sirius Red staining protocol as described previously (*Pattabiraman et al., 2015*). Briefly, the sections were deparaffinized, rehydrated by ethanol gradient and deionized water, then incubated in Picro Sirius Red Solution (ab150681; Abcam, Cambridge, MA, USA) for 60 min. After rinsing the sections quickly in acetic acid solution (ab150681; Abcam, Cambridge, MA, USA), an ethanol dehydration was performed. Pictures were taken with a conventional light microscope using the above settings. The staining intensity of the TM region was scored by a masked reviewer (YD) on a scale from 1 to 4.

### Statistics

Data were presented as the mean $\pm$ standard error and analyzed by PASW 18.0 (SPSS Inc., Chicago, IL, USA). One-way ANOVA was performed to compare IOPs among the different groups while paired $t$ test was used for ingroup comparison to each baseline. The Kruskal–Wallis and Mann–Whitney $U$ test were used to compare the grading of ECM staining. A statistical difference of $p < 0.05$ was considered significant. A linear mixed effects model was fitted to the fold change response in R (*R Core Team, 2016*; *Dang et al., 2017b*). The response was modeled as a B-spline function of time with 5 degrees of freedom (*Berk, 2013* ; *Hu et al., 1998*).

## RESULTS

### Trabecular meshwork morphology, histology and ECM assessment

Two eyes per group were discarded due to leaks while the baseline was established. In eyes that were successfully cultured, the gross morphology of the anterior chambers remained well preserved after two freeze-thaw cycles, with light opacification of the corneas as the most notable change (Fig. 1). Histology from within 24 h after exposure to freeze-thaw (F) or saponin (S) indicated that F preserved the microarchitecture better Figs. 2A and 2B than

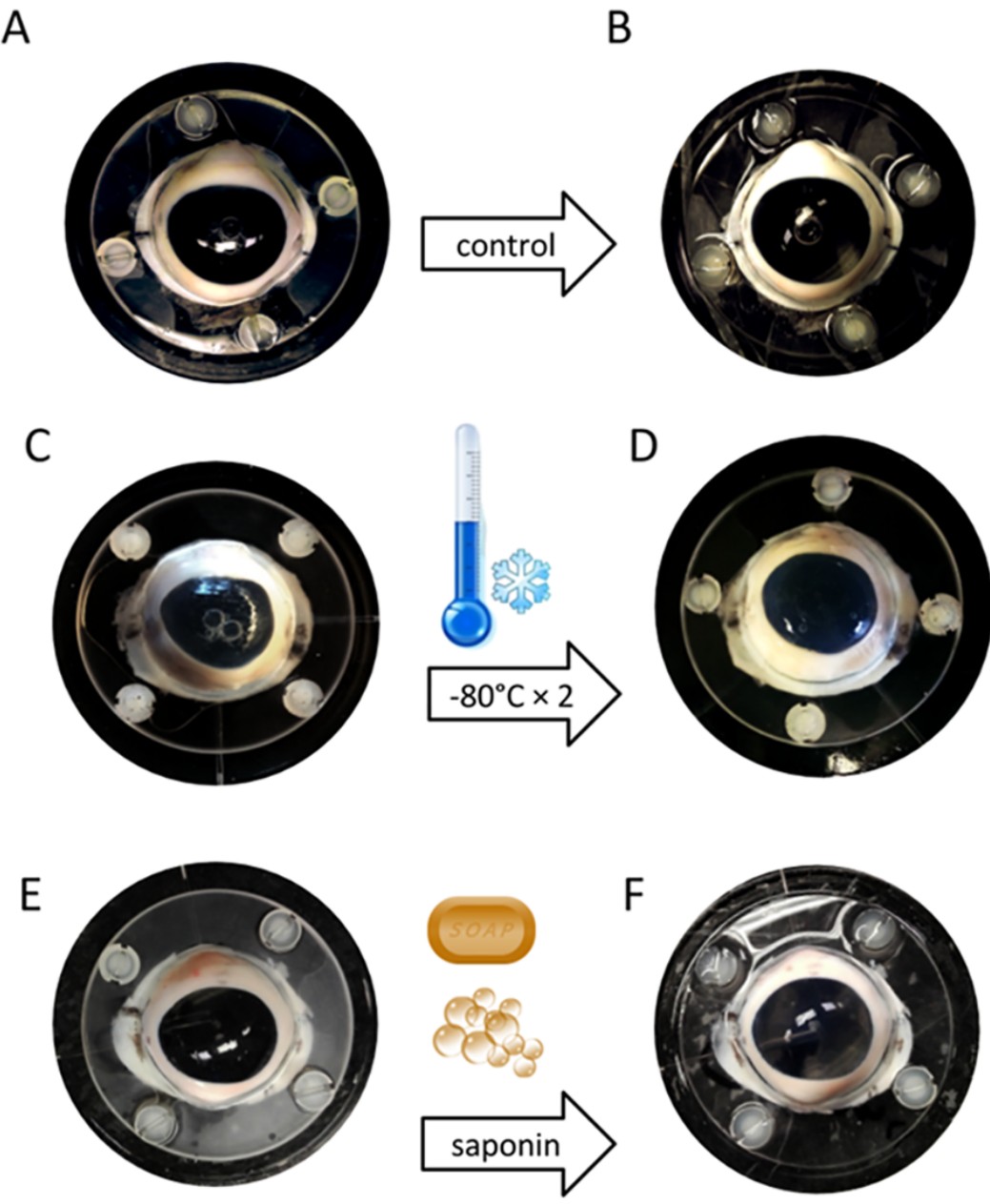

**Figure 1** **Freeze-thaw treatment of anterior segment cultures.** Eyes were exposed to two cycles of freezing at −80 °C followed by thawing at room temperature. Compared to the eyes with sham treatment (A and B), the macroscopic appearance of the eyes after freeze-thaw cycles (C and D) or saponin treatment (E and F) remained mostly unchanged.

S (Fig. 2C). Blue stained nucleoli could still be observed but disappeared later, consistent with the viability assay results described below. We assessed the TM-ECM (*Cormack, 2001*; *Fischer et al., 2008*) by Picro Sirius Red staining. Compared to the normal control (Fig. 3A), neither freeze-thaw cycles (Fig. 3B) nor saponin (Fig. 3C) caused a significant loss or increase of the total ECM deposition after 180 h perfusion ($P = 0.324$ and $P = 0.095$, respectively; Fig. 3D).

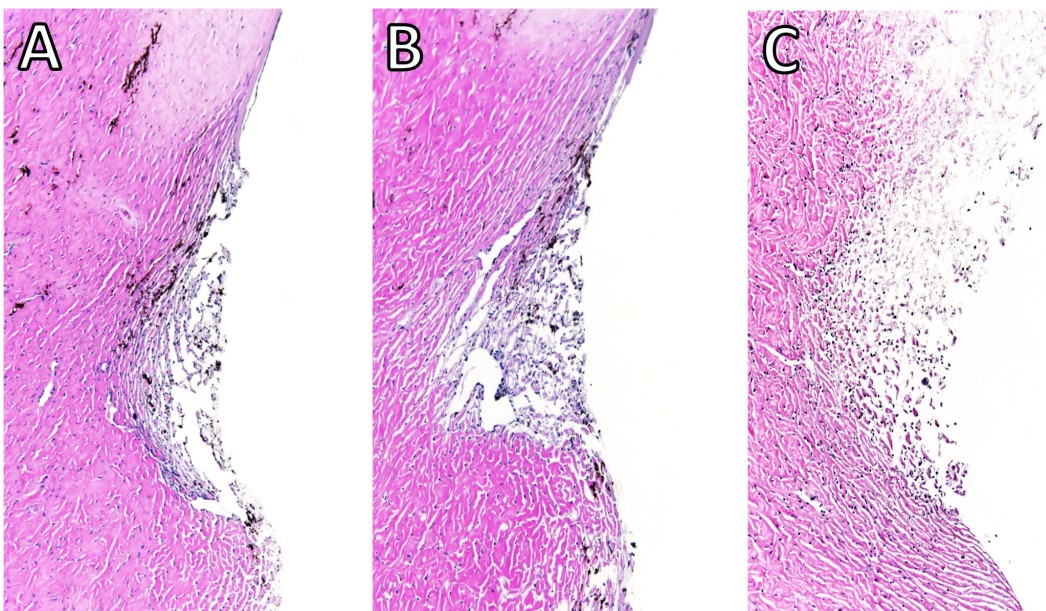

**Figure 2  Histology of the angle of perfused anterior chambers.** Control eyes (A) had an appearance that was similar to that of freeze-thaw treated eyes (B). Saponin-treated eyes (C) presented with a less compact structure. Blue nuclei could be seen in all sections.

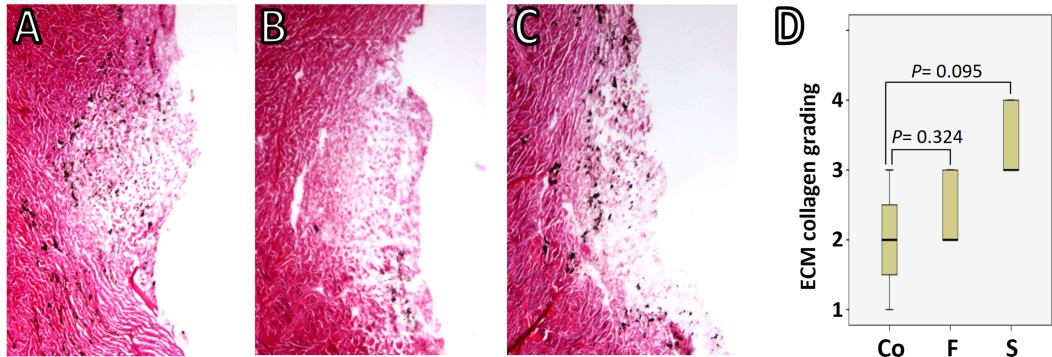

**Figure 3  TM-ECM assessment by collagen staining.** Total collagen was stained with Picro Sirius Red and scored by a masked observer (Yalong Dang) from 1 to 4 according to the staining intensity. Collagen deposition in the normal control (A) is comparable to that in either freeze-thaw group (B) or saponin-treated group (C) after 180 h perfusion ($P = 0.324$ and $P = 0.095$, respectively) (D).

## Monitoring of TM ablation

Ablation control eyes were transduced with a relatively low titer of $1 \times 10^7$ eGFP FIV vectors prior to F and S. 24 h after transduction, the TM cells began to express eGFP, reaching a peak intensity at 48 h, as reported previously (*Loewen et al., 2016e*; *Dang et al., 2017d*). There were discontinuous areas of transduced TM (Figs. 4A, 4B, 4E and 4F) and transduction along corneal stretch folds as well as sclera. Two cycles of $-80$ °C completely abolished eGFP fluorescence Figs. 4C and 4D. Two cycles were necessary because pilot eyes with only one cycle still showed some eGFP positive cells. In contrast, after 0.02% saponin

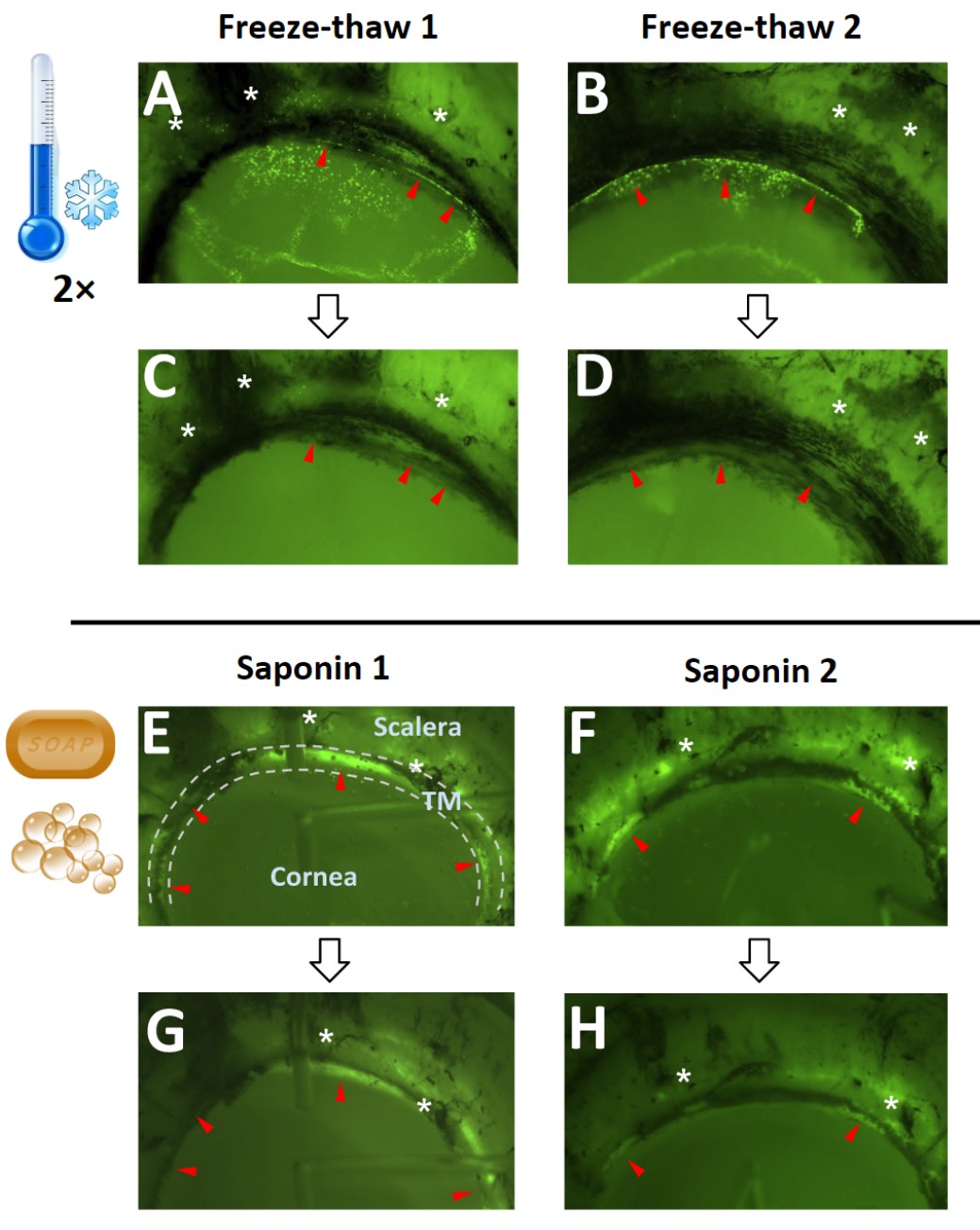

**Figure 4** **Confirmation of cytoablation.** EGFP expression of GINSIN transduced cells (A and B) vanished after two freeze-thaw cycles (C and D). In contrast, eGFP could still be seen in saponin-treated eyes (G and H). Red arrowheads point to transduced trabecular meshwork (TM) (between dashed lines) that was ablated after freeze-thaw. Sclera, cornea and TM are labeled in the first saponin ablated eye (E). The black lines are pigmented areas of the scleral spur and sclera where the ciliary body attached. Asterisks are placed near landmarks to make it easier to compare the before and after treatment images.

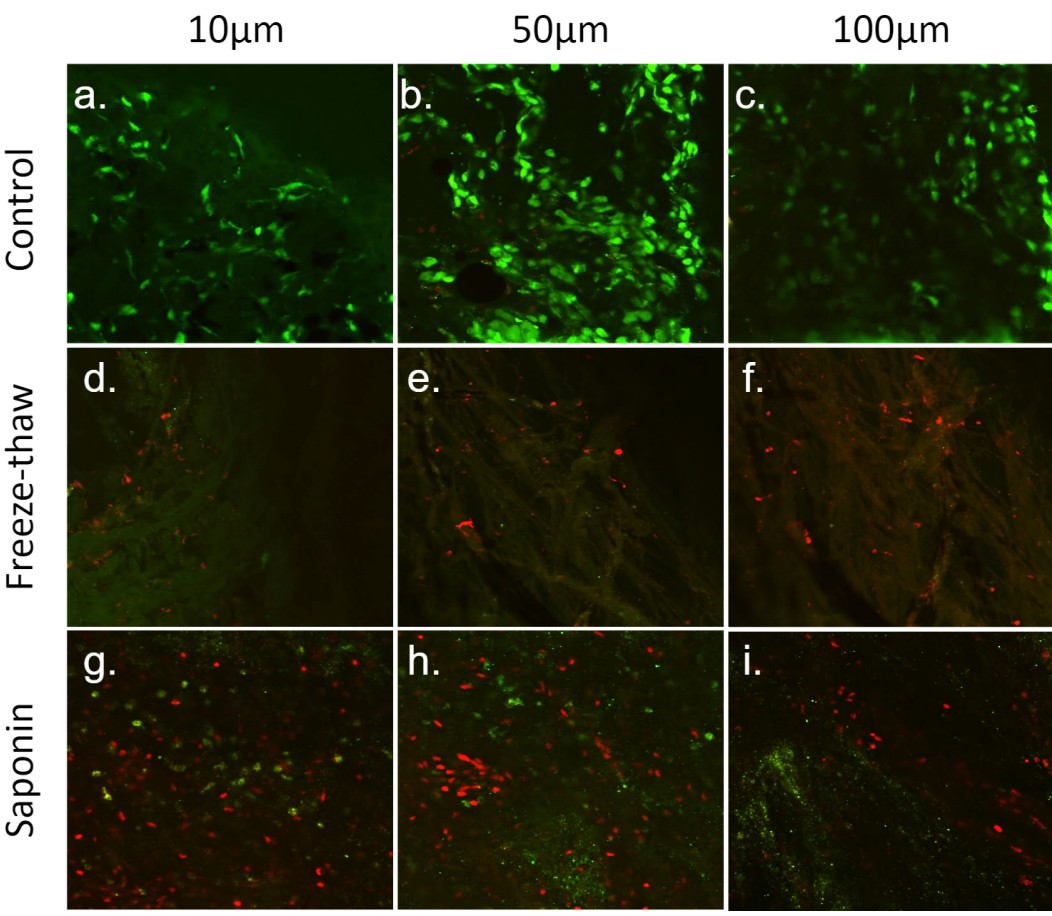

**Figure 5** **Assessment of TM cell viability by calcein AM/PI co-labelling.** Viable trabecular meshwork (TM) cells exposed to calcein AM showed bright green fluorescence, while dead TM cells allowed PI to enter cell membrane and label the cell nuclear with red fluorescence. In the control group, most TM cells were still viable after perfusion for two weeks (A–C). In contrast, cells, including many nuclei, were destroyed by freeze-thaw. No calcein AM, and only a few PI-labeled TM cells were found (D–F). Different from the other two groups, a few TM cells were still alive in the saponin-treated group, but most of them were labeled as dead cells by PI (G–I). The different microscopy depths are approximate samples from the uveal (A), corneoscleral (D) and cribriform layer (G) in this flat mount.

perfusion, eGFP fluorescence appeared quenched, and only a small portion of transduced cells was ablated 24 h after exposure (Figs. 4G and 4H).

## TM viability

Calcein AM and PI staining were used as a viability assay to validate TM ablation by freeze-thaw or saponin exposure. Most TM cells from the control group were positive with calcein AM showing bright green fluorescence (Figs. 5A–5C), while only occasionally stained with PI (Figs. 5B and 5C) at the TM depths of 10 $\mu$m, 50 $\mu$m and 100 $\mu$m. In contrast, no calcein AM staining and very few PI-stained cells were found in the freeze-thaw group (Figs. 5D–5F). Different from the above two groups, most of the TM cells in the saponin group were labeled by PI, with few cells demonstrating a light calcein AM staining (Figs. 5G and 5H).

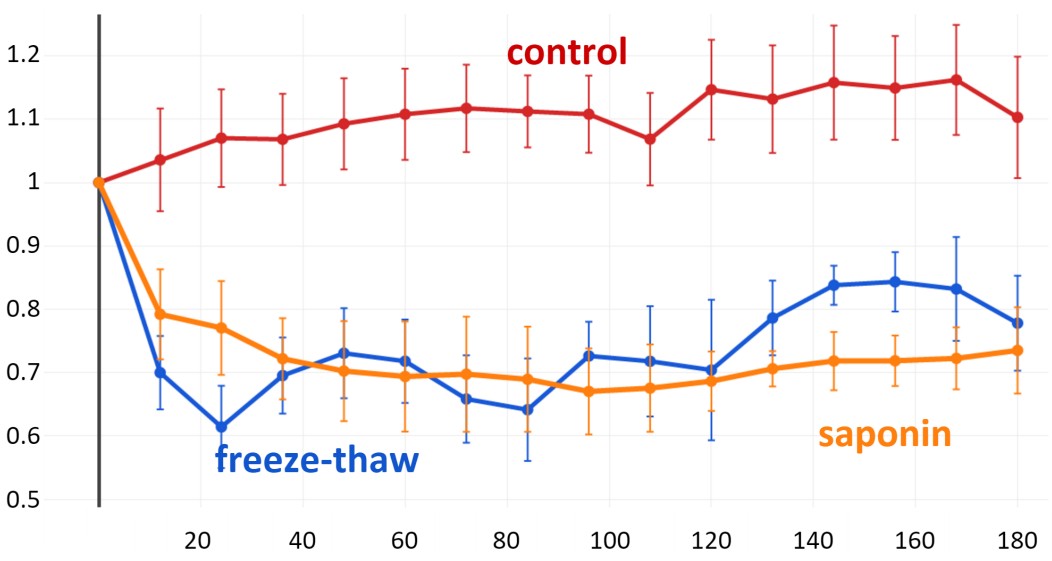

**Figure 6** **IOP Reduction after TM decellularization.** Freeze-thaw (F) resulted in a more rapid IOP reduction than saponin (S) (averages ± SEM). There were no differences at any single time between F and S. Differences between controls and S were not significant onward from 96 h.

## IOPs

A stable baseline was established for all anterior segments for 72 h before exposure to F or S. IOPs varied insignificantly by 10.3 ± 7.5% throughout the end of the study (all $P > 0.05$ compared to the baseline) (Fig. 6). However, pressure decreased dramatically after either freeze-thaw or saponin (baseline freeze-thaw 14.75 ± 2.24 mmHg, baseline saponin 14.37 ± 1.14 mmHg, $P = 0.288$). At 12 h, F dropped to 70 ± 7.1% and S to 79.2 ± 8.1% from the baseline, respectively. F remained significantly lower than C for 96 h ($P = 0.02$), eyes experienced a larger IOP variability onward resulting in reduced significance. In contrast, S had a significantly lower IOP throughout the study until the experimental endpoint at 180 h. We applied a linear mixed effects model that used a B-spline function of time with 5 degrees of freedom (*Berk, 2013*) (Supplemental Information 1). The results reflect the averages shown in Fig. 5 and confirm the three non-linear behaviors with distinctly different patterns. F had an intercept, representative of the initial IOP drop, that was −0.378 fold less ($p < 0.001$) than C and a standard error of 0.088 with 15 degrees of freedom and a $t$-value of −4.3. F was not significantly different from S in the B-spline function model ($p = 0.142$). S had an intercept that was 0.242 fold less than C ($p = 0.013$) with a standard error of 0.086 and 15 degrees of freedom.

## DISCUSSION

In this study, we developed a method to decellularize the trabecular meshwork in anterior segment perfusion cultures quickly and reliably. This was achieved with two cycles of freezing at −80 °C and thawing at room temperature. This avoids the use of chemical agents that might dissolve the ECM or have other, not well-characterized effects. We compared this method to a saponin-mediated ablation. Each method has distinct properties

and advantages. Freeze-thaw cycles, applied here to group F, have been used extensively before to ablate tissues in treatment of human diseases (*Erinjeri & Clark, 2010*; *Baust et al., 2014*; *Chu & Dupuy, 2014*) including cyclocryodestruction in glaucoma (*Benson & Nelson, 1990*). It has also been used in research (*Baust et al., 2014*; *Chan & Ooi, 2016*; *Liu et al., 2016*) and in food production (*US Food and Drug Administration, 2011*; *Gill, 2006*; *Craig, 2012*). Mechanisms of cryoablation in medicine include direct cell injury, vascular injury, ischemia, apoptosis, and immunomodulation (*Chu & Dupuy, 2014*): cell injury during freezing causes dehydration from the so-called solution effect that causes the earlier freezing extracellular compartment to extract solutes, an osmotic gradient and cell shrinkage (*Lovelock, 1953*). This can be enhanced by ice crystal formation within the cell, damaging organelles and the cell membrane. During thawing, the intracellular compartment shifts to hypertonia, attracting fluid that causes the cell to burst. Mechanisms not at work in our model are direct cold-induced coagulative necrosis that is the result of sublethal temperatures that activate apoptosis (*Baust & Gage, 2005*) and direct, cold-induced coagulative necrosis from vascular injury as a result of stasis, thrombosis, and ischemia. An interesting clinical effect is the intense inflammation after cryoablation that is different from heat coagulation as immunogenic epitopes are preserved (*Jansen et al., 2010*).

Saponin, used in experimental group S, can lyse cells. At lower concentrations, it has also been used to reduce the viability of cells (*Abu-Hassan et al., 2015*). Saponins are an enormously large class of chemical compounds that exist in a range of plant species (Saponaria) which can produce soap-like foam when shaken in aqueous solution and has been used in early soaps (*Coombes, 2012*). These substances are amphiphilic (both hydro- and lipophilic) glycosides in which a sugar is bound to a functional three-terpene group via a glycosidic bond. Saraponins are an important subset of saponins that are steroidal while aglycone derivatives have the pharmacologic characteristics of alkaloids. Historically, saponins have been used as a poison for fishing (*Campbell, 1999*). The ability of saponins to form complexes with cholesterol has been exploited for both therapeutic and research purposes. These cholesterol-saponin complexes create pores in the cell membrane, which can enhance the penetration of macromolecules but also induce lysis (*Holmes et al., 2015*). All of the above properties may have wide-ranging and difficult to characterize effects in cell transplantation models. Additionally, different batches of saponin may have a different composition of compounds. It is, therefore, necessary to determine the proper concentration for saponin with different lot numbers to increase the reproducibility of experiments.

The macroscopic appearance of group F compared to group S samples had only minor differences which consisted of mild opacification of the cornea. The microscopic architecture was well preserved in F, but less so in S. This can be expected based on the properties of these two different methods described above. Especially the change of permeability of cell membranes by saponin might cause edema by allowing fluids to enter the extracellular space more easily compared to freeze-thaw that is more likely to cause dehydration (*Mazur, 1963*; *Mazur, Rall & Leibo, 1984*; *Wolkers et al., 2007*). Compared to the cells themselves, many blue nuclei persisted in early histology because they are less permeable and contain less fluid compared to the cytoplasm. These observations were

reflected in the ablation of transduced, eGFP expressing cells. Freeze-thaw caused nearly complete loss of fluorescence after the first cycle and disappeared entirely when cells were disrupted after the second cycle. Saponin appears to have caused leakage of eGFP proteins were diminished fluorescence was observed, but only a few cells were fully lysed.

The results of TM viability assay by calcein AM/PI staining further confirmed the findings from the histological analysis and eGFP ablation. Freeze-thaw caused the disappearance of almost all cells secondary to the above mechanism of cell dehydration and subsequent burst. Saponin appears to have been a sublethal injury to many cells, especially in the uveal and corneoscleral TM. *Abu-Hassan et al. (2015)* developed a protocol to induce a sublethal injury with saponin to mimic a glaucomatous TM injury in an *ex vivo* model and were able to correct the glaucoma phenotype. We observed a modest decline in a model of inducible cytoablation mediated by an HSV-tk suicide vector (*Zhang et al., 2014*).

This pattern of cell death matches the IOP reduction of groups F and S. F experienced a more immediate drop compared to S as could be expected by a complete breakdown of the TM outflow regulation. In comparison, the slower downslope seen after saponin exposure likely reflects the more gradual cell function decline with eventual cell death. The final IOP was lower in S which may represent the loss not only of cells but also of hydrophilic components of the ECM which could persist in eyes in F to a longer extent and time. H&E and Picro Sirius red, used here to obtain a comprehensive characterization of the ECM, do not differentiate between its two main categories, proteoglycans and fibrous proteins, or their subgroups (*Frantz, Stewart & Weaver, 2010*). More than the fibrous scaffold of the ECM, hydrogel-like ECM components require a continuous production and have a more fleeting nature. Because of this, timing of of cell seeding in repopulation experiments will be crucial.

Our use of a B-spline function provides a more appropriate description of effects in an eye culture model that play out over a period rather than the common comparison of single time points. Single time point comparisons assume incorrectly that observations are largely unrelated (*Hu et al., 1998*). Handling longitudinal data with B-spline functions extends the standard linear mixed-effects models and accounts for a broad range of non-linear behaviors (*Dang et al., 2017b*). B-spline functions are robust to small sample sizes, as well as to noisy observations and missing data.

Consistent with our clinical (*Akil et al., 2016*; *Loewen et al., 2016d*; *Neiweem et al., 2016*; *Dang et al., 2016b*; *Dang et al., 2016a*; *Kaplowitz et al., 2016*; *Roy et al., 2017*) and laboratory findings (*Zhang et al., 2014*; *Parikh et al., 2016a*; *Wang et al., 2017*), TM ablation improves outflow and lowers IOP. A 20.8 ± 8.1% IOP reduction was achieved at 12 h after saponin treatment and a 30.0 ± 7.1% reduction in the freeze-thaw group. The freeze-thaw cycles removed all meshwork cells, including those in the corneoscleral and cribriform TM which account for at least 50% of trabecular outflow resistance, whereas most of these cells were preserved in the saponin group. It is possible that the IOP reduction seen after cyclocryodestruction is at least partially due to an improvement of conventional outflow and not only due to reduced aqueous humor production or uveoscleral outflow enhancement from inflammation (*Gorsler, Thieme & Meltendorf, 2015*).

The limitations of this study are that cytoablation via freeze-thaw may liberate other, undesirable factors from non-trabecular cells that also die. The argument against a profound impact of those is that the macroscopic and microscopic structures were surprisingly stable for the entire time of 10 days. We only describe an ablation method here but not a repopulation of the trabecular meshwork by cell transplantation. Transduction (*Loewen et al., 2001*; *Loewen et al., 2002*; *Loewen et al., 2016f*) affects high outflow areas more than low flow areas of the eye (*Loewen et al., 2016b*; *Loewen et al., 2016c*; *Parikh et al., 2016b*; *Dang et al., 2017c*) which is the result of a higher multiplicity of infection (m.o.i., transduction events per cell). We speculate that these areas might also be more affected by saponin but equally ablated by freeze-thaw cycles.

In conclusion, we developed a fast, inexpensive and reliable method that results in complete ablation of TM cells while the architecture was well-preserved.

### Funding
This work was supported by the National Eye Institute (K08-EY022737), the Initiative to Cure Glaucoma of the Eye and Ear Foundation of Pittsburgh and the Wiegand Fellowship, Eye and Ear Foundation of Pittsburgh and an unrestricted fellowship grant from the Third Xiangya Hospital of Central South University. The funders had no role in study design, data collection and analysis, decision to publish, or preparation of the manuscript.

### Grant Disclosures
The following grant information was disclosed by the authors:
National Eye Institute: K08-EY022737.
Initiative to Cure Glaucoma of the Eye and Ear Foundation of Pittsburgh and the Wiegand Fellowship.
Eye and Ear Foundation of Pittsburgh.
Third Xiangya Hospital of Central South University.

### Competing Interests
The authors declare there are no competing interests.

### Author Contributions
- Yalong Dang conceived and designed the experiments, performed the experiments, analyzed the data, wrote the paper, prepared figures and/or tables, reviewed drafts of the paper.
- Susannah Waxman and Chao Wang performed the experiments, analyzed the data, reviewed drafts of the paper.
- Adrianna Jensen performed the experiments, reviewed drafts of the paper.
- Ralitsa T. Loewen performed the experiments, contributed reagents/materials/analysis tools, prepared figures and/or tables, reviewed drafts of the paper.
- Richard A. Bilonick performed the experiments, analyzed the data, contributed reagents/materials/analysis tools, reviewed drafts of the paper.

# PeerJ

- Nils A. Loewen conceived and designed the experiments, analyzed the data, contributed reagents/materials/analysis tools, wrote the paper, prepared figures and/or tables, reviewed drafts of the paper.

## Data Availability

The raw data has been supplied as a Supplementary File.

## Supplemental Information

Supplemental information for this article can be found online at http://dx.doi.org/10.7717/peerj.3629#supplemental-information.

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
