# Peer review of "Freeze-thaw decellularization of the trabecular meshwork in an ex vivo eye perfusion model"

_PeerJ, doi:10.7717/peerj.3629_

## Round 0.1 · original submission · Major Revisions

· Academic Editor

Major Revisions

Dear author,

The reviews were generally favorable and we look forward to recieving your revisions.

Reviewer 1 ·

Basic reporting

This manuscript comports well with each of the basic reporting criteria although it is not a hypothesis drive analysis.

Experimental design

Research question- can improved cellular ablation result from freeze thaw cycles vs. chemical treatment of a biological matrix? Study contains appropriate controls, assay methodologies and can be replicated.

Validity of the findings

Data is sound and support the conclusions.

Reviewer 2 ·

Basic reporting

The English language should be upgraded for clarity. Some words are used incorrectly, such as “substrate”, there are many missing words, and there is often a disagreement in plurality between the subject and verb. Please have your manuscript edited by a native English speaking colleague. Many, but not all words or phrases of concern have been highlighted or have a sticky note attached to them as examples to be corrected.

The introduction/background is good, and explains current knowledge of the field quite well, with sufficient literature references.

The figures are mixed in quality. Fig. 2- There is a typo in the caption. Fig. 3-The authors need to label the trabecular meshwork, sclera, and cornea. Where were the images taken, in high, low, or intermediate flow areas? What is the black line near the eGFP? Fig. 4-It is unclear how the authors can see to the cribriform layer with the confocal, unless they did frontal sections. The confocal microscope can only go 50 microns deep, while the cribriform layer is about 150 microns deep. What is figure 4? Is this tissue? How can the authors see the cribriform? Fig. 5- It would help to label the x and y axis of the graph. Fig. 6-It is not clear that Fig. 6 is necessary. If the data needs elaborate statistics, it would be better to increase the n to establish validity. Raw data supplied as PeerJ policy.

Experimental design

The research question is well-defined, but it needs to be stated how research fills an identified knowledge gap, and why this is worthwhile doing. There needs to be some clarification of the use of confocal microscopy to visualize the cribriform area, as it does not have the capacity to investigate that deep into the TM. This would be better visualized with some type of EM or by using frontal sections.

Validity of the findings

The authors need to better explain their use of the B-spline, which does not add to the readability of the text, and it is unclear why they need to use it for ANOVA. Did they do multi-testing correction?

Where there is speculation in the discussion, this should be stated as such.

Additional comments

The authors need to consider the effect of segmental flow with saponin treatment. This treatment will only flow saponin into high flow regions, so if only 2 quadrants are taken to examine at random for eGFP, it is possible that they could get 2 high flow areas, 2 low flow areas, or a mix to the two. What does this do to repeatability if the n is low?

Miscellaneous points:

A. Saponin flowed in goes only to the high flow regions of the TM.
B. Freeze-thaw should affect all the TM cells.
C. The microscopic pictures shown for control, freeze-thaw, and saponin-treatment need to be at a much higher resolution. It is not possible to examine ECM for damage at this resolution, so does not bolster the claim of ECM damage by one or the other treatment.

Annotated reviews are not available for download in order to protect the identity of reviewers who chose to remain anonymous.

·

Basic reporting

This manuscript is well written and the findings are clearly described in the text.

Experimental design

Within the scope of the questions posed, the experimental paradigms used are satisfactory.
1. Yet, histological examinations of extracellular matrix can enhance this manuscript. Since the authors mention that the extracellular matrix materials are retained after freeze thaw decellularlization but they fail to demonstrate it. Addition of immunostaining and electron microscopy study to demonstrate the loss of cellularity but retainment of extracellular matrix will enhance the manuscript.
2. Also it is important to remember that the extracellular matrix is an important signaling molecule and mechanotransducers bound to integrins. Therefore, since the authors claim that there are no cells or cellular machinery, they have to explain how does the extracellular matrix material stay there.

Validity of the findings

The findings are interesting and this technique will definitely have an impact in the field.

Additional comments

It would be interesting to see if stem cells can be made to repopulate these ablated spaces as an option for cellular therapy.

---

## Round 0.2 · accepted · Accept

· Academic Editor

Accept

Dear Dr. Loewen,

I accept your revisions and I believe that the paper is now acceptable for publication.